# Generalized Semi-Supervised Learning via Self-Supervised Feature Adaptation

**Jiachen Liang**[1,2], **Ruibing Hou**[1], **Hong Chang**[1,2], **Bingpeng Ma**[2], **Shiguang Shan**[1,2], **Xilin Chen**[1,2]
[1] Institute of Computing Technology, Chinese Academy of Sciences
[2]University of Chinese Academy of Sciences
jiachen.liang@vipl.ict.ac.cn, {houruibing, changhong, sgshan, xlchen}@ict.ac.cn, bpma@ucas.ac.cn

## Abstract

Traditional semi-supervised learning (SSL) assumes that the feature distributions of labeled and unlabeled data are consistent which rarely holds in realistic scenarios. In this paper, we propose a novel SSL setting, where unlabeled samples are drawn from a mixed distribution that deviates from the feature distribution of labeled samples. Under this setting, previous SSL methods tend to predict wrong pseudo-labels with the model fitted on labeled data, resulting in noise accumulation. To tackle this issue, we propose *Self-Supervised Feature Adaptation* (SSFA), a generic framework for improving SSL performance when labeled and unlabeled data come from different distributions. SSFA decouples the prediction of pseudo-labels from the current model to improve the quality of pseudo-labels. Particularly, SSFA incorporates a self-supervised task into the SSL framework and uses it to adapt the feature extractor of the model to the unlabeled data. In this way, the extracted features better fit the distribution of unlabeled data, thereby generating high-quality pseudo-labels. Extensive experiments show that our proposed SSFA is applicable to various pseudo-label-based SSL learners and significantly improves performance in labeled, unlabeled, and even unseen distributions.

## 1   Introduction

Semi-Supervised Learning (SSL) uses a small amount of labeled data and a large amount of unlabeled data to alleviate the pressure of data labeling and improve the generalization ability of the model. Traditional SSL methods [21, 43, 32, 28, 22, 30, 5, 6] usually assume that the feature distribution of unlabeled data is consistent with that of labeled one. However, in many real scenarios, this assumption may not hold due to different data sources. When unlabeled data is sampled from distributions different from labeled data, traditional SSL algorithms will suffer from severe performance degradation, which greatly limits their practical application.

In real-world scenarios, it is quite common to observe feature distribution mismatch between labeled and unlabeled samples. *On the one hand, unlabeled samples could contain various corruptions.* For example, in automatic driving, the annotated images used for training can hardly cover all driving scenes, and a plethora of images under varying weather and camera conditions are captured during real driving. Similarly, in medical diagnosis, individual differences and shooting conditions among patients could incur various disturbances in unlabeled data. *On the other hand, unlabeled samples could contain unseen styles.* For example, in many tasks, the labeled samples are typically real-world photos, while unlabeled data collected from the Internet usually contain more styles that are not present in the labeled data, such as cartoons or sketches. Notably, although the distributions of these unlabeled data may differ from the labeled data, there is implicit common knowledge between them that can compensate for the diversity and quantity of training data. Therefore, it is crucial to enlarge the SSL scope to effectively utilize unlabeled data from different distributions.

37th Conference on Neural Information Processing Systems (NeurIPS 2023).

In this study, we focus on a more realistic scenario of Feature Distribution Mismatch SSL (FDM-SSL), *i.e.*, the feature distributions of labeled and unlabeled data could be different and the feature distributions of test data could contain multiple distributions. It is generally observed that the performance of classical SSL algorithms [22, 30, 6, 9] degrades substantially under feature distribution mismatch. Specifically, in the early stages of training, classical SSL algorithms typically utilize the current model fitted on labeled data to produce pseudo-labels for unlabeled data. However, when the distribution of unlabeled data deviates, the current model are not applicable to unlabeled data, resulting in massive incorrect pseudo-labels and aggravating confirmation bias [3]. Recently, some works [7, 19] attempt to address FDM-SSL. These approaches assume that the unlabeled feature distribution comes from a single source and they only focus on the labeled distribution, which do not always hold in real tasks. However, due to the unknown test scenarios, it is desirable to develop a method to perform well on labeled, unlabeled and even unseen distributions simultaneously.

In this work, we propose a generalized *Self-Supervised Feature Adaptation* (SSFA) framework for FDM-SSL which does not need to know the distribution of unlabeled data ahead of time. The core idea of SSFA is to decouple pseudo-label predictions from the current model to address distribution mismatch. SSFA consists of two modules, including the semi-supervised learning module and the feature adaptation module. Inspired by [31] that the main classification task can be indirectly optimized through the auxiliary task, SSFA incorporates an auxiliary self-supervised task into the SSL module to train with the main task. In the feature adaptation module, given the current model primarily fitted on labeled data, SSFA utilizes the self-supervised task to update the feature extractor before making predictions on the unlabeled data. After the feature extractor adapts to the unlabeled distribution, refined features can be used to generate more accurate pseudo-labels to assist SSL.

Furthermore, the standard evaluation protocol of SSL normally assumes that test samples follow the same feature distribution as labeled training data. However, this is too restricted to reflect the diversity of real-world applications, where different tasks may focus on different test distributions. It is strongly desired that the SSL model can perform well across a wide range of test distributions. Therefore, in this work, we propose new evaluation protocols that involve test data from labeled, unlabeled and unseen distributions, allowing for a more comprehensive assessment of SSL performance. Extensive experiments are conducted on two types of feature distribution mismatch SSL tasks, *i.e.*, corruption and style mismatch. The experimental results demonstrate that various pseudo-label-based SSL methods can be directly incorporated into SSFA, yielding consistent performance gains across a wide range of test distributions.

## 2   Related work

**Semi-Supervised Learning (SSL).** Existing SSL methods [44, 7, 30, 41, 50, 9, 6] typically combine various commonly used techniques for semi-supervised tasks, such as consistency regularization[21, 43, 32, 28] and pseudo-label [22], to achieve state-of-the-art performance. For example, [6] expands the dataset by interpolating between labeled and unlabeled data. [5, 30] use the confident weak augmented view prediction results to generate pseudo-labels for the corresponding strong augmented view. [41, 50, 9, 44] improve the performance by applying adaptive thresholds instead of fixed thresholds. However, these works focus on traditional semi-supervised learning, which assumes that the labeled data and unlabeled data are sampled from the same distribution. Recently, some works [2, 53] address the issue of different feature distributions for labeled and unlabeled data under certain prerequisites. [2] assumes the test and unlabeled samples are drawn from the same single distribution. [53] studies Semi-Supervised Domain Generalization (SSDG), which requires multi-source partially-labeled training data. To further expand SSL to a more realistic scenario, we introduce a new Feature Distribution Mismatch SSL (FDM-SSL) setting, where the unlabeled data comes from multiple distributions and may differ from the labeled distribution. To solve the FDM-SSL, we propose Self-Supervised Feature Adaptation (SSFA), a generic framework to improve the performance in labeled, unlabeled, and even unseen test distributions.

**Unsupervised Domain Adaptation (UDA).** UDA aims to transfer knowledge from a source domain with sufficient labeled data to an unlabeled target domain through adaptive learning. The UDA methods can be roughly categorized into two categories, namely metric-based methods [23, 35, 25, 34, 49] and adversarial methods [1, 16, 8, 24, 46, 29, 37, 10, 42]. Metric-based methods focus on measuring domain discrepancy to align the feature distributions between source and unlabeled target domains. Adversarial methods [16, 47, 24] use a domain classifier as a discriminator to enforce the

Table 1: Comparison between different problem settings.

| Task | Labeled | Unlabeled | Train setting | Test distribution |
|---|---|---|---|---|
| traditional SSL | scarce | abundant | $p_l(x) = p_u(x)$ | $p_l(x)$ |
| UDA | abundant | abundant | $p_l(x) \neq p_u(x)$ | $p_u(x)$ |
| TTA | abundant | - | $p_l(x)$ | $p_u(x)$ |
| FDM-SSL | scarce | abundant | $p_l(x) \neq p_u(x)$ | $p_l(x), p_u(x), p_{unseen}(x)$ |

feature extractor to learn domain-invariant features through adversarial training. Another task similar to FDM-SSL is Unsupervised Domain Expansion (UDE) [39, 51, 33, 45], which aims to maintain the model's performance on the labeled domain after adapting to the unlabeled domain. The main differences between UDA, UDE and FDM-SSL are the number of labeled data during training and the distribution of the unlabeled data. In FDM-SSL, the labeled data is scarce, and the distribution of unlabeled data is not limited to a specific domain but rather a mixture of multiple domains. These challenges make UDA and UDE methods unable to be directly applied to FDM-SSL.

**Test-Time Adaptation (TTA).** If the distribution of test data differs from that of training data, the model's performance may suffer performance degradation. TTA methods focus on improving the model's performance with a two-stage process: a model should first adapt to the test samples and then make predictions of them. For example, [31] and [15] introduce an auxiliary self-learning task to adapt the model to test distribution. [38] modifies the scaling and bias parameters of BatchNorm layers based on entropy minimization. [52, 13, 12, 26] try to improve the robustness of test-time adaptation. Unlike most TTA methods which suppose the test samples come from a single distribution, [40] assumes a mixed test distribution that changes continuously and stably. However, these assumptions are still difficult to satisfy in real-life scenarios. In addition, similar to UDA, TTA methods generally require a model trained well on the source domain. In contrast, FDM-SSL setting only requires scarce labeled data and has no restrictions on the distribution of unlabeled data.

## 3 Problem Setting

For Feature Distribution Mismatch SSL (FDM-SSL) problem, a model observes a labeled set $D_l = \{(x_i, y_i)\}_{i=1}^{N_l}$ and an unlabeled set $D_u = \{(u_j)\}_{j=1}^{N_u}$ with $N_u \gg N_l$, where $x_i$ and $u_i$ are input images and $y_i$ is the label of $x_i$. The labeled image $x_i$ and unlabeled image $u_j$ are drawn from two different feature distributions $p_l(x)$ and $p_u(x)$, respectively. Note that different from previous works, in FDM-SSL setting the unlabeled samples may come from a mixture of multiple distributions rather than just one distribution: *i.e.* $p_u(x) = w_0 p_l(x) + \sum_{k=1}^{K} w_k p_u^k(x)$, where $w_0$ and $w_k$ represent the weights of the labeled distribution $p_l(x)$ and the $k-$th unlabeled distribution $p_u^k(x)$ respectively. The goal of FDM-SSL is to train a model that generalizes well over a large range of varying test data distributions, including labeled, unlabeled and even distributions unseen during training ($p_{unseen}(x)$). The differences between traditional SSL, UDA, TTA and FDM-SSL, are summarized in Table 1.

The core of FDM-SSL is to enhance the utilization of unlabeled samples in the presence of feature distribution mismatch. On one hand, the shared information, such as patterns and structures, in the unlabeled samples can provide effective cues for model adaptation. On the other hand, these mismatched unlabeled samples can facilitate the learning of a more robust model by exposing it to a wide range of data distributions.

## 4 Method

### 4.1 Overview

In the FDM-SSL setting, due to the feature distribution shift from $p_l(x)$ to $p_u(x)$, directly applying the SSL model fitted on labeled data to unlabeled data may lead to massive inaccurate predictions on the unlabeled data, thereby aggravating confirmation bias and impairing SSL learning.

This motivates us to propose Self-Supervised Feature Adaptation (SSFA), a unified framework for FDM-SSL, which decouples the pseudo-label predictions from the current model to address distribution mismatch. As illustrated in Figure 1, SSFA consists of two modules: a semi-supervised

learning module and a feature adaptation module. The semi-supervised learning module incorporates a pseudo-label-based SSL learner with a self-supervised *auxiliary* task that shares a portion of feature extractor parameters with the *main* classification task. [15] has pointed out that the main task can be indirectly optimized by optimizing the auxiliary task. To this end, the feature adaptation module is designed to update the current model through self-supervised learning on unlabeled data, so as to better match the unlabeled distribution and predict higher-quality pseudo-labels for semi-supervised learning. Therefore, by optimizing the auxiliary self-supervised task individually, the updated model can make more accurate classification decisions in the main task.

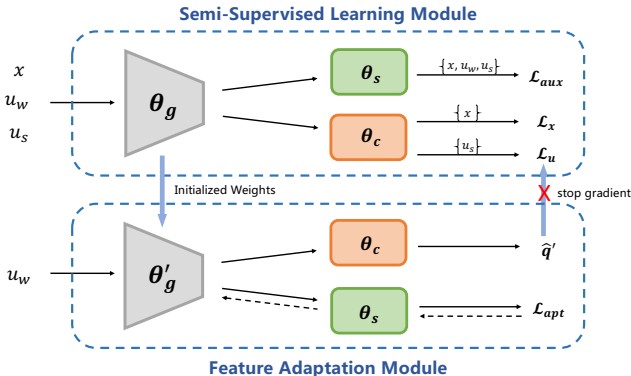

Figure 1: The pipeline of SSFA. Let $x$, $u_w$ and $u_s$ denote a batch of the labeled data, the weak augmentation and the strong augmentation of unlabeled data respectively, $\{\cdot\}$ represent the data stream.

## 4.2 Semi-Supervised Learning Module

In the Semi-Supervised Learning Module, we introduce a self-supervised task as an auxiliary task, which is optimized together with the main task. As shown in Figure 1, the network parameter $\theta$ comprises three parts: $\theta_g$ for the shared encoder, $\theta_c$ for the main task head, and $\theta_s$ for the auxiliary task head. During training, SSL module optimizes a *supervised loss* $\mathcal{L}_x$, an *unsupervised loss* $\mathcal{L}_u$ and a *self-supervised auxiliary loss* $\mathcal{L}_{aux}$, simultaneously. Typically, given a batch of labeled data $\{(x_b, y_b)\}_{b=1}^{B}$ with size $B$ and a batch of unlabeled data $\{(u_b)\}_{b=1}^{\mu B}$ with size $\mu B$, where $\mu$ is the ratio of unlabeled data to labeled data, $\mathcal{L}_x$ applies standard cross-entropy loss on labeled examples:

$$\mathcal{L}_x = \frac{1}{B} \sum_{b=1}^{B} \mathrm{H}(x_b, y_b; \theta_g, \theta_c), \tag{1}$$

where $\mathrm{H}(\cdot, \cdot)$ is the cross-entropy function. Different SSL learners [30, 5] may design different $\mathcal{L}_u$. One of the widely used is the pseudo-label loss. In particular, given the pseudo-label $q_b$ for each unlabeled input $u_b$, $\mathcal{L}_u$ in traditional SSL can be formulated as:

$$\mathcal{L}_u = \frac{1}{\mu B} \sum_{b=1}^{\mu B} \Omega(u_b, q_b; \theta_g, \theta_c), \tag{2}$$

where $\Omega$ is the per-sample supervised loss, *e.g.*, mean-square error [6] and cross-entropy loss [30].

Furthermore, we introduce a self-supervised learning task to the SSL module for joint training to optimize the parameter of the auxiliary task head $\theta_s$. To learn from labeled and unlabeled data distributions simultaneously, it is necessary to use all samples from both distributions for training. Therefore, $\mathcal{L}_{aux}$ can be formulated as:

$$\mathcal{L}_{aux} = \frac{1}{(2\mu+1)B} \left( \sum_{b=1}^{B} \ell_s(x_b; \theta_g, \theta_s) + \sum_{b=1}^{\mu B} \ell_s(\mathcal{A}_w(u_b); \theta_g, \theta_s) + \sum_{b=1}^{\mu B} \ell_s(\mathcal{A}_s(u_b); \theta_g, \theta_s) \right), \tag{3}$$

where $\ell_s$ denotes the self-supervised loss function, $\mathcal{A}_w$ and $\mathcal{A}_s$ denote the weak and strong augmentation functions respectively. In order to maintain consistent optimization between the main task

and auxiliary task, it is necessary to optimize both tasks jointly. So we add $\mathcal{L}_{aux}$ to semi-supervised learning module and the final object is:

$$\mathcal{L}_{\text{SSFA}} = \mathcal{L}_x + \lambda_u \mathcal{L}_u + \lambda_a \mathcal{L}_{aux}, \tag{4}$$

where $\lambda_u$ and $\lambda_a$ are hyper-parameters denoting the relative weights of $\mathcal{L}_u$ and $\mathcal{L}_{aux}$ respectively.

### 4.3 Feature Adaptation Module

Some classic SSL approaches [30, 6, 5] directly use the outputs of the current classifier as pseudo-labels. In particular, [30] applies weak and strong augmentations to unlabeled samples and generates pseudo-labels using the model's predictions on weakly augmented unlabeled samples, *i.e.*, $q_b = p_m(y|\mathcal{A}_w(u_b); \theta_g, \theta_c)$, where $p_m(y|u; \theta)$ denotes the predicted class distribution of the model $\theta$ given unlabeled image $u$. However, as the classification model $(\theta_g, \theta_c)$ is mainly fitted to the labeled distribution, the prediction on $\mathcal{A}_w(u_b)$ is usually inaccurate under distribution mismatch between the labeled and unlabeled samples.

To alleviate this problem, we design a Feature Adaptation Module to adapt the model to the unlabeled data distribution before making predictions, thereby producing more reliable pseudo-labels for optimizing the unsupervised loss $\mathcal{L}_u$. More specifically, before making pseudo-label predictions, we firstly fine-tune the shared feature extractor $\theta_g$ by minimizing the self-supervised auxiliary task loss on unlabeled samples:

$$\mathcal{L}_{apt} = \frac{1}{\mu B} \sum_{b=1}^{\mu B} \ell_s(\mathcal{A}_w(u_b); \theta_g, \theta_s). \tag{5}$$

Here $\theta_g$ is updated to $\theta_g' = \arg\min \mathcal{L}_{apt}$. Notably, since excessive adaptation may lead to deviation in the optimization direction and largely increase calculation costs, we only perform one-step optimization in the adaptation stage.

After self-supervised adaptation, we use $(\theta_g', \theta_c)$ to generate the updated prediction, which can be denoted as:

$$q_b' = p_m(y|\mathcal{A}_w(u_b); \theta_g', \theta_c). \tag{6}$$

We convert $q_b'$ to the hard "one-hot" label $\hat{q}_b'$ and denote $\hat{q}_b'$ as the pseudo-label for the corresponding strongly augmented unlabeled sample $\mathcal{A}_w(u_b)$. After that, $\theta_g'$ will be discarded without influencing other model parts during training. In the end, $\mathcal{L}_u$ in the SSL module (Equation 4) can then be computed with cross-entropy loss as:

$$\mathcal{L}_u = \frac{1}{\mu B} \sum_{b=1}^{\mu B} \mathbb{I}(\max(q_b') > \tau) \mathrm{H}(\mathcal{A}_s(u_b), \hat{q}_b'; \theta_g, \theta_c). \tag{7}$$

### 4.4 Theoretical Insights

In our framework, we jointly train the model with a self-supervised task with loss function $\ell_s$, and a main task with loss function $\ell_m$. Let $h \in \mathcal{H}$ be a feasible hypothesis and $D_u = \{(u_i, y_i^u)\}_{i=1}^N$ be the unlabeled dataset. Note that the ground-truth label $y_i^u$ of $u_i$ is actually unavailable, but just used for analysis.

**Lemma 1 ([31])** *Assume that for all $x, y$, $\ell_m(x, y; h)$ is differentiable, convex and $\beta$-smooth in $h$, and both $\|\nabla \ell_m(x, y; h)\|, \|\nabla \ell_s(x; h)\| \leq G$ for all $h \in \mathcal{H}$. With a fixed learning rate $\eta = \frac{\epsilon}{\beta G^2}$, for every $x, y$ such that $\langle \nabla \ell_m(x, y; h), \nabla \ell_s(x; h) \rangle > \epsilon$, we have*

$$\ell_m(x, y; h') < \ell_m(x, y; h), \tag{8}$$

*where $h'$ is the updated hypothesis, namely $h' = h - \eta \nabla \ell_s(x, y; h)$.*

Let $\hat{E}_m(h, D_u) = \frac{1}{K} \sum_{i=1}^K \ell_m(x_u^i, y_u^i; h)$ denote the empirical risk of the main task on $D_u$, and $\hat{E}_s(h, D_u) = \frac{1}{K} \sum_{i=1}^K \ell_s(x_u^i; h)$ denote the empirical risk of the self-supervised task on $D_u$. Under the premise of Lemma 1 and the following sufficient condition, with a fixed learning rate $\eta$:

$$\langle \nabla \hat{E}_m(h, D_u), \nabla \hat{E}_s(h, D_u) \rangle > \epsilon, \tag{9}$$

we can get that:

$$\hat{E}_m(h', D_u) < \hat{E}_m(h, D_u), \tag{10}$$

where $h'$ is the updated hypothesis, namely $h' = h - \eta\nabla\hat{E}_s(h, D_u)$.

Therefore, in the smooth and convex case, the empirical risk of the main task $\hat{E}_m$ can theoretically tend to 0 by optimizing the empirical risk of self-supervised task $\hat{E}_s$. Thus, in our feature adaptation module, by optimizing the self-supervised loss for unlabeled samples, we can indirectly optimize the main loss, thereby mitigating confirmation bias and making full use of the unlabeled samples. As above analysis, the gradient correlation plays a determining factor in the success of optimizing $\hat{E}_m$ through $\hat{E}_s$ in the smooth and convex case. For non-convex deep loss functions, we provide empirical evidence to show that our theoretical insights also hold. Figure 2 plots the correlation between the gradient inner product (of the main and auxiliary tasks) and the performance improvement of the model on the

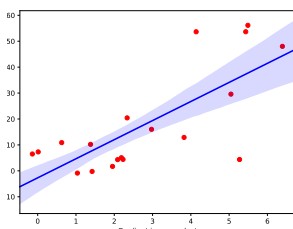

Figure 2: Scatter plot of the gradient inner product between the two tasks, and the improvement from SSFA. We transform the x-axis with $\log(x) + 1$ for clarity.

test set, where each point in the figure represents the average result of a set of test samples. In Figure 2, we observe that there is a positive correlation between the gradient inner product and model performance improvement for non-convex loss functions on the deep learning model. This phenomenon is consistent with the theoretical conclusion, that is, a stronger gradient correlation indicates a higher performance improvement.

## 5 Experiments

In this section, we evaluate our proposed SSFA framework on various datasets, where labeled and unlabeled data are sampled from different distributions. To this end, we consider two different distribution mismatch scenarios: image corruption and style change, in the following experiments. More experimental details and results are provided in the Appendix.

### 5.1 Experimental Setting

**Datasets.** For the image corruption experiments, we create the labeled domain by sampling images from CIFAR100 [20], and the unlabeled domain by sampling images from a mixed dataset consisting of CIFAR100 and CIFAR100 with Corruptions (CIFAR100-C). The CIFAR100-C is constructed by applying various corruptions to the original images in CIFAR100, following the methodology used for ImageNet-C [18]. We use ten types of corruption as training corruptions while reserving the other five corruptions as unseen corruptions for testing. The proportion of unlabeled samples from CIFAR100-C was controlled by the hyper-parameter $ratio$. For the style change experiments, we use OFFICE-31 [27] and OFFICE-HOME [36] benchmarks. Specifically, we designate one domain in each dataset as the labeled domain, while the unlabeled domain comprises either another domain or a mixture of multiple domains.

**Implementation Details.** In corruption experiments, we use WRN-28-8 [48] as the backbone except for [6] where WRN-28-2 is used to prevent training collapse. In style experiments, we use ResNet-50 [17] pre-trained on ImageNet [14] as the backbone for OFFICE-31 and OFFICE-HOME. To ensure fairness, we use the same hyperparameters for different methods employed in our experiments.

**Comparison Methods.** We compare our method with three groups of methods: (1) Supervised method as a baseline to show the effectiveness of training with unlabeled data.(2) Classical UDA methods including DANN [1] and CDAN [24]; and (3) Popular SSL methods including MixMatch [6], ReMixMatch[5], FixMatch [30], FM-Rot (joint training with rotation prediction task), FM-Pre (pre-trained by rotation prediction tasks), FreeMatch [41], SoftMatch [9], and Adamatch [7].

**Evaluation Protocols.** We evaluate the performance across labeled, unlabeled and unseen distributions to verify the general applicability of our framework. Therefore, we consider three evaluation protocols: (1) **Label-Domain Evaluation** (L): test samples are drawn from *labeled* distribution, (2)

Table 2: Comparison of accuracy (%) for Feature Distribution Mismatch SSL on CIFAR100.

| Method | 400 labeled | | | | | | 4000 labeled | | | 10000 labeled | | |
|---|---|---|---|---|---|---|---|---|---|---|---|---|
| | $ratio$ 0.5 | | | $ratio$ 1.0 | | | $ratio$ 1.0 | | | $ratio$ 1.0 | | |
| | L | UL | US | L | UL | US | L | UL | US | L | UL | US |
| Supervised | 10.6 | 8.9 | 6.9 | 10.6 | 8.9 | 6.9 | 48.0 | 17.0 | 24.0 | 61.6 | 20.8 | 30.8 |
| DANN [1] | 11.3 | 9.3 | 7.0 | 11.4 | 9.3 | 7.0 | 46.7 | 30.8 | 27.6 | 61.4 | 42.7 | 37.8 |
| CDAN [24] | 11.6 | 9.5 | 7.3 | 11.6 | 10.0 | 7.5 | 47.1 | 31.3 | 28.8 | 62.6 | 41.1 | 39.1 |
| MixMatch [6] | 10.7 | 3.6 | 5.4 | 15.4 | 2.5 | 6.1 | 46.4 | 10.9 | 21.1 | 60.3 | 31.4 | 37.6 |
| AdaMatch [7] | 6.8 | 4.6 | 1.3 | 6.0 | 4.5 | 2.1 | 19.1 | 6.4 | 1.8 | 26.7 | 9.0 | 1.6 |
| ReMixMatch [5] | 39.6 | 20.6 | 32.7 | 39.2 | 19.1 | 3.1 | 68.4 | 35.8 | 58.1 | 75.5 | 40.2 | 63.2 |
| RM-SSFA (ours) | **43.1** | **22.9** | **35.1** | **43.0** | **23.1** | **35.1** | **71.0** | **37.9** | **60.8** | **76.4** | **41.6** | **63.5** |
| FixMatch [30] | 25.8 | 4.7 | 16.5 | 15.7 | 3.5 | 8.5 | 53.0 | 16.0 | 39.0 | 65.3 | 33.0 | 52.8 |
| FM-SSFA (ours) | **37.0** | **23.2** | **25.8** | **25.7** | **22.2** | **22.5** | **60.2** | **52.5** | **51.2** | **69.1** | **57.8** | **58.0** |
| FreeMatch [41] | 35.0 | 1.4 | 21.0 | 17.4 | 2.2 | 7.2 | 55.2 | 16.3 | 41.4 | 67.0 | 23.9 | 53.4 |
| FreeM-SSFA (ours) | **37.6** | **25.1** | **31.4** | **27.5** | **18.7** | **21.9** | **62.0** | **54.2** | **53.0** | **70.2** | **61.8** | **60.0** |
| SoftMatch [9] | 35.5 | 2.9 | 24.1 | 19.4 | 4.6 | 12.8 | 58.3 | 29.7 | 48.9 | 68.3 | 34.7 | 56.3 |
| SM-SSFA (ours) | **40.8** | **29.5** | **34.8** | **31.1** | **22.4** | **25.2** | **62.6** | **54.4** | **53.8** | **70.3** | **61.8** | **59.9** |

**UnLabel-Domain Evaluation** (UL): test samples are drawn from *unlabeled* distribution, and (3) **UnSeen-Domain Evaluation** (US): test samples are drawn from *unseen* distribution.

## 5.2 Main Results

**Image Corruption.** Our SSFA framework is a generic framework that can be easily integrated with existing pseudo-label-based SSL methods. In this work, we combine SSFA with four SSL methods: FixMatch, ReMixMatch, FreeMatch and SoftMatch, denoted by FM-SSFA, RM-SSFA, FreeM-SSFA and SM-SSFA, respectively. Table 2 shows the compared results on the image corruption experiment with different numbers of labeled samples and $ratio$. We can observe that (1) the UDA methods, DANN and CDAN, exhibit very poor performance on labeled-domain evaluation (L). This is because UDA methods aim to adapt the model to the unlabeled domain, which sacrifices performance on the labeled domain. And the UDA methods are primarily designed to adapt to a *single* unlabeled distribution, making it unsuitable for FDM-SSL scenarios. In contrast, our methods largely outperform the two UDA methods on all evaluations. (2) The traditional SSL methods suffer from significant performance degradation in the FDM-SSL setting, particularly in unlabeled-domain evaluation (UL). And this degradation becomes more severe when the number of labeled data is small and the proportion of unlabeled data with corruption is high, due to the increased feature mismatch degree. Our methods largely outperform these SSL methods, validating the effectiveness of our self-supervised feature adaptation strategy for FDM-SSL. (3) SSFA largely improves the performance of original SSL methods on all three evaluations. The gains on unlabeled-domain evaluation (UL) indicate that SSFA can generate more accurate pseudo-labels for unlabeled samples and reduce confirmation bias. Moreover, the gains on unseen-domain evaluation (US) validate that SSFA can enhance the model's robustness and generalization.

**Style Change.** We conduct experiments on OFFICE-31 and OFFICE-HOME benchmarks to evaluate the impact of style change on SSL methods. Specifically, we evaluate SSL methods in two scenarios where unlabeled data is sampled from a single distribution and a mixed distribution. The results are summarized in Table 3. Compared to the supervised method, most SSL methods present significant performance degradation. One possible explanation is that style change causes an even greater feature distribution mismatch compared to image corruption, resulting in more erroneous predictions for unlabeled data. As shown in Table 3(a), our method shows consistent performance improvement over existing UDA and SSL methods on OFFICE-31. In some cases, such as "W/A", the task itself may be relatively simple, so using only label data can already achieve high accuracy. The results on OFFICE-HOME are summarized in Table 3(b). To demonstrate that our method can indeed improve the accuracy of pseudo-labels, we add an evaluation protocol **UU**, which indicates the *pseudo-labels accuracy* of selected unlabeled samples. As shown in Table 3(b), compared to existing UDA and SSL methods, our approach achieves the best performance on all setups and significantly improves the accuracy of pseudo-labels on unlabeled data. The results provide experimental evidence that supports the theoretical analysis in Section 4.4.

Table 3: Comparison of accuracy (%) for Feature Distribution Mismatch SSL on OFFICE-31 and OFFICE-HOME.

(a) Results on OFFICE-31

| Method | Single Domain | | | | | | | | | | | | Multiple Domains | | | |
| | A/D | | | A/W | | | D/A | | | W/A | | | A/DW | | D/AW | |
| | L | UL | US | L | UL | US | L | UL | US | L | UL | US | L | UL | L | UL |
|---|---|---|---|---|---|---|---|---|---|---|---|---|---|---|---|---|
| supervised | 65.9 | 57.1 | 52.0 | 65.9 | 47.8 | 61.0 | **93.0** | 56.8 | 84.9 | **93.6** | 56.2 | 92.4 | 68.0 | 51.9 | **93.3** | 60.7 |
| DANN [1] | 53.9 | 17.9 | 20.1 | 15.3 | 7.2 | 4.6 | 74.7 | 4.8 | 16.7 | 2.5 | 2.9 | 4.2 | 28.1 | 7.6 | 4.7 | 2.8 |
| CDAN [24] | 2.9 | 5.6 | 2.6 | 2.9 | 2.3 | 4.2 | 4.7 | 2.9 | 2.6 | 41.3 | 7.7 | 18.5 | 2.9 | 3.2 | 45.5 | 10.7 |
| FixMatch [30] | 58.4 | 46.9 | 53.3 | 56.3 | 9.5 | 19.3 | 82.8 | 4.7 | 50.6 | 78.3 | 9.4 | 62.6 | 55.0 | 48.8 | 80.2 | 40.3 |
| FM-Rot | 60.0 | 53.1 | 57.1 | 54.6 | 27.4 | 19.3 | 70.8 | 29.5 | 50.6 | 75.9 | 17.9 | 66.9 | 54.0 | 40.1 | 63.0 | 26.9 |
| FM-SSFA | **67.5** | **61.7** | **63.0** | **66.4** | **56.5** | **62.5** | **93.0** | **59.3** | **85.9** | 93.0 | **56.4** | 92.8 | **69.1** | **60.8** | 92.4 | **61.8** |

(b) Results on OFFICE-HOME

| Method | A/CPR | | | C/APR | | | P/ACR | | | A/ACPR | | | C/ACPR | | | R/ACPR | | |
| | L | UL | UU | L | UL | UU | L | UL | UU | L | UL | UU | L | UL | UU | L | UL | UU |
|---|---|---|---|---|---|---|---|---|---|---|---|---|---|---|---|---|---|---|
| supervised | 48.4 | 39.9 | - | 40.9 | 31.2 | - | 68.4 | 36.5 | - | 49.3 | 43.0 | - | 41.0 | 36.3 | - | 61.6 | 46.8 | - |
| DANN [1] | 45.0 | 37.8 | - | 31.8 | 7.7 | - | 53.9 | 25.6 | - | 49.3 | 42.0 | - | 30.5 | 20.2 | - | 44.5 | 32.1 | - |
| CDAN [24] | 30.4 | 22.8 | - | 1.2 | 1.8 | - | 63.0 | 34.3 | - | 4.4 | 4.2 | - | 10.5 | 6.9 | - | 13.5 | 10.2 | - |
| FixMatch [30] | 27.2 | 17.7 | 19.3 | 28.1 | 19.6 | 19.4 | 55.6 | 24.9 | 36.0 | 32.4 | 23.0 | 31.0 | 36.9 | 30.6 | 36.0 | 42.2 | 31.5 | 35.4 |
| FM-Pre | 21.2 | 15.0 | 15.4 | 28.6 | 22.9 | 25.4 | 51.9 | 22.0 | 26.3 | 30.6 | 20.1 | 24.8 | 37.9 | 33.8 | 41.7 | 33.4 | 23.8 | 24.9 |
| FM-Rot | 1.7 | 2.1 | 0.0 | 33.4 | 20.1 | 0.7 | 43.4 | 14.6 | 17.2 | 25.6 | 17.7 | 21.3 | 37.3 | 31.0 | 33.2 | 45.6 | 34.7 | 42.3 |
| FM-SSFA | **50.7** | **44.0** | **69.0** | **45.1** | **37.6** | **66.2** | **70.6** | **41.9** | **70.5** | **55.0** | **45.5** | **73.2** | **44.7** | **41.7** | **66.0** | **64.8** | **52.7** | **80.3** |

## 5.3 Feature Visualization

Figure 3 visualizes the domain-level features generated by SSL models with/without SSFA respectively. In Figure 3 (a), the vanilla FixMatch model maps labeled and unlabeled samples to different clusters in the feature spaces without fusing samples from different domains. Conversely, Figure 3 (b) shows that our FM-SSFA model can effectively fuse these samples. We further compute the $A$-distance metric [4] to measure the distributional divergence between the labeled and unlabeled distributions. The $A$-distance of FM-SSFA (0.91) is lower than that of FixMatch (1.10), verifying the implicit distribution alignment of SSFA.

Additionally, Figure 4 visualizes the class-level features generated by SSL models with/without SSFA respectively. In Figure 4 (a), FixMatch fails to distinguish samples from different classes. In contrast, FM-SSFA can separate samples from different categories well, as shown in Figure 4 (b). This result indicates that SSFA can help to extract more discriminative features.

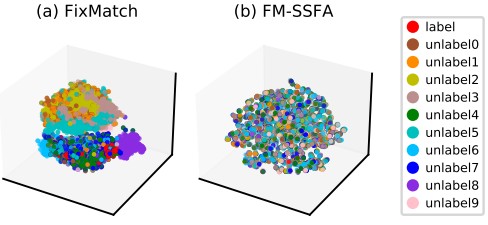

Figure 3: Visualization of domain-level features using different methods, where "label" represents the labeled data drawn from the labeled domain, and "unlabel0" to "unlabel9" represent the unlabeled data drawn from ten unlabeled domains respectively.

Figure 4: Visualization of class-level features using different methods.

## 5.4 Ablation Study

**Combined with different self-supervised tasks.** Table 4 compares different self-supervised tasks combined with our SSFA framework on the image corruption experiment. We employ different self-supervised losses corresponding to the rotation prediction task, contrastive learning task, and

Table 4: Combined with different self-supervised tasks. "Rot", "SimClr", and "EM" represent the rotation prediction task, contrastive learning task [11] and entropy minimization task [38] respectively.

| Method | 400 labeled | | | | | | 4000 labeled | | | 10000 labeled | | |
|---|---|---|---|---|---|---|---|---|---|---|---|---|
| | $ratio$ 0.5 | | | $ratio$ 1.0 | | | $ratio$ 1.0 | | | $ratio$ 1.0 | | |
| | L | UL | US | L | UL | US | L | UL | US | L | UL | US |
| FixMatch[30] | 25.8 | 4.7 | 16.5 | 15.7 | 3.5 | 8.5 | 53.0 | 16.0 | 39.0 | 65.3 | 33.0 | 52.8 |
| FM-SSFA (Rot) | **37.0** | **23.2** | **25.8** | **25.7** | **22.2** | **22.5** | **60.2** | **52.5** | **51.2** | **69.1** | **57.8** | **58.0** |
| FM-SSFA (SimClr) | 22.8 | 13.7 | 17.4 | 19.2 | 13.6 | 14.1 | 55.3 | 43.8 | 45.1 | 66.2 | 53.6 | 52.9 |
| FM-SSFA (EM) | 22.9 | 14.5 | 17.2 | 18.0 | 15.9 | 14.6 | 55.2 | 44.2 | 44.7 | 66.7 | 52.5 | 52.6 |

entropy minimization task, which are the cross entropy loss, the contrastive loss from SimCLR and the entropy loss, respectively. As shown, different self-supervised tasks can bring significant performance gains. And the rotation prediction task leads to the largest improvements compared to the other two self-supervised tasks. This can be attributed to the more optimal parameters and the direct supervision signals in rotation prediction. In the experiments, we employ the rotation prediction task as the default self-supervised task.

**The effectiveness of Feature Adaptation Module.** To evaluate the effectiveness of the Feature Adaptation Module in the SSFA framework, we compare FM-SSFA with a baseline method FM-Rot, where we remove the Feature Adaptation Module and only add the self-supervised rotation prediction task. As shown in Table 3(a) and 3(b), FM-SSFA largely outperforms FM-Rot on OFFICE-31 and OFFICE-HOME, especially in the UL evaluation metric. The superiority of FM-SSFA highlights that the Feature Adaptation Module helps the model to adapt to unlabeled samples from different distributions, resulting in better generalization on unlabeled domain. Moreover, we can observe the FM-Rot is only marginally better than FixMatch and, in some cases, may even perform worse. These results suggest that simply integrating the self-supervised task into SSL methods only brings limited performance gains and may even be detrimental in some scenarios.

**Small distribution shift between labeled and unlabeled data.** To demonstrate the robustness of our proposed method, we evaluate SSFA on scenarios where there is a small distribution shift between labeled and unlabeled data. Specially, we conduct experiments on two setups: $ratio$=0.0 (degrade into traditional SSL) and $ratio$=0.1. Table 5 shows our method can still bring significant improvements over the baseline. This indicates that our proposed SSFA framework is robust to various SSL scenarios even under a low noise ratio.

Table 5: The performance of SSFA when the distribution shift between labeled and unlabeled data is small on CIFAR100.

| Method | $ratio$ 0.0 | | $ratio$ 0.1 | | |
|---|---|---|---|---|---|
| | L/UL | US | L | UL | US |
| FixMatch [30] | 33.3 | 25.8 | 31.7 | 10.5 | 24.8 |
| FM-SSFA | **41.3** | **33.0** | **41.2** | **15.8** | **32.9** |

**Robustness to confident threshold $\tau$.** Figure 5 illustrates the impact of confident threshold $\tau$ for different SSL models on OFFICE-31. As shown in Figure 5, the vanilla FixMatch model is highly sensitive to the values of $\tau$. For example, when $\tau$ is set to 0.95, FixMatch achieves relatively high performance with confident enough pseudo-labels. When $\tau$ is lowered to 0.85, the performance of FixMatch on UL and UU degrades significantly, indicating that the model has deteriorated by too many wrong pseudo-labels. In contrast, FM-SSFA is more robust to different values of $\tau$.

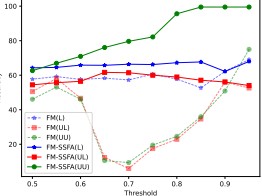

Figure 5: The impact of $\tau$ for different SSL models on OFFICE-31 ("A/W" task).

**The number of shared layers between auxiliary and main task.** Table 6 analyzes the impact of different numbers of shared layers in the shared feature extractor of the auxiliary and main task. As shown, the performance difference is negligible between sharing 2 and 3 layers in the feature extractor, while a significant performance degradation happens if we share all layers (4 layers) of the feature extractor. We argue that this is because too many shared parameters of the feature extractor may lead to over-adaptation of the feature extractor and compromise of the main task, thus the predictions of

Table 6: The impact of shared layers between auxiliary and main task on OFFICE-HOME. "X layers" denotes the number of shared layers for the feature extractor.

| Method | A/CPR | | | C/APR | | | P/ACR | | | R/ACP | | |
|---|---|---|---|---|---|---|---|---|---|---|---|---|
| | L | UL | UU | L | UL | UU | L | UL | UU | L | UL | UU |
| FM-SSFA (2 layers) | **48.9** | **39.9** | 70.8 | **46.0** | **39.5** | 69.0 | 70.3 | 42.0 | 67.3 | 61.0 | 40.8 | 67.0 |
| FM-SSFA (3 layers) | 47.5 | 39.8 | **71.0** | 45.4 | 39.0 | **77.6** | **72.3** | **45.4** | **74.4** | **63.4** | **43.9** | **76.8** |
| FM-SSFA (4 layers) | 44.4 | 39.1 | 57.1 | 1.3 | 2.4 | 0.0 | 57.9 | 33.4 | 59.5 | 40.3 | 28.7 | 54.3 |

pseudo-labels may be more erroneous. In the experiments, we set the number of shared layers to 2 by default.

## 6   Conclusion

In this paper, we focus on a realistic SSL setting, FDM-SSL, involving a mismatch between the labeled and unlabeled distributions, complex mixed unlabeled distributions and widely unknown test distributions. The primary challenge lies in the scarcity of labeled data and the potential presence of mixed distributions within the unlabeled data. To address this challenge, we propose a generalized framework SSFA, which introduces a self-supervised task to adapt the feature extractor to the unlabeled distribution. By incorporating this self-supervised adaptation, the model can improve the accuracy of pseudo-labels to alleviate confirmation bias, thereby enhancing the generalization and robustness of the SSL model under distribution mismatch.

**Broader Impacts and Limitations.** The new problem setting takes into account the model's generalization across different distributions, which is essential for expanding the application of SSL methods in real-world scenarios. Additionally, SSFA serves as a simple yet effective framework that can be seamlessly integrated with any pseudo-label-based SSL methods, enhancing the overall performance and adaptability of these models. However, the performance of SSFA is affected by the shared parameters between the main task and the auxiliary task. We hope that SSFA can attract more future attention to explore the effectiveness of feature adaptation with the self-supervised task.

## Acknowledgments

This work is partially supported by National Key R&D Program of China no. 2021ZD0111901, National Natural Science Foundation of China (NSFC): 61976203, 62276246 and 62306301, and National Postdoctoral Program for Innovative Talents under Grant BX20220310.

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
