# OpenReview forum: "Generalized Semi-Supervised Learning via Self-Supervised Feature Adaptation"
_NeurIPS.cc/2023/Conference — NeurIPS 2023 poster_

### Official Review · Reviewer_Pxab · 2023-07-02

**Soundness:** 3 good
**Presentation:** 3 good
**Contribution:** 2 fair
**Rating:** 5
**Confidence:** 3

**Summary:**

The authors claim to have addressed the new problem of semi-supervised learning with feature distribution mismatch (FDM-SSL) and to have proposed a new method called Self-Supervised Feature Adaptation (SSFA) for this problem.

The proposed SSFA consists of semi-supervised learning module and a feature adaptation module. The semi-supervised learning module optimizes the classifier based on joint minimization of supervised/unsupervised losses (to pseudo-labels) and a self-supervised auxiliary loss (for rotation prediction) with weak/strong data augmentation. The feature adaptation module aims to generate more reliable pseudo labels by updating the feature extraction backbone based on the self-supervised auxiliary loss on the unlabeled samples.

Comparative experiments with existing semi-supervised learning are conducted and the effectiveness of the proposed method is claimed.

**Strengths:**

This paper addresses the FDM-SSL problem, which has few studied examples.

The method is simple and its superiority over the major semi-supervised learning methods in the FDM-SSL problem is experimentally demonstrated.

**Weaknesses:**

A. Problem Novelty

The authors claim in the abstract and conclusion sections that one of the novelties of this paper is in that they tackle the novel problem of semi-supervised learning with feature distribution mismatch. However, this claim is not valid because previous work [a] has addressed a very similar problem. The differences between these two should be emphasized and clarified.

Furthermore, there are also several closely related problems that have been considered in the past (e.g., semi-supervised domain generalization [b] and few-shot domain generalization [c]), but discussion of their relevance is lacking.

[a] Aminian et al., An Information-theoretical Approach to Semi-supervised Learning
under Covariate-shift. ICML, 2022.

[b] Zhou et al., Semi-Supervised Domain Generalization with Stochastic StyleMatch. NeurIPS Workshop, 2021. (Extended version is published at IJCV)

[c] Yuan et al., A novel forget-update module for few-shot domain generalization. Pattern Recognition, 2022.


B. Experiments

Comparisons with existing methods for the related problems I listed in Weakness A above are missing. Furthermore, since the problem of FDM-SSL can be considered a variant of domain generalization, there should be comparisons with existing domain generalization (and more recent domain adaptation) methods. Due to the lack of such an evaluation, the effectiveness of the proposed method is not adequately demonstrated.


C. Method Novelty

The novelty of the proposed method is lacking. It is basically composed of common techniques in semi-supervised learning (such as weak/strong data augmentation or pseudo labeling with confidence thresholding). The feature adaptation module, which performs fine-tuning of the feature extraction backbone based on self-supervised auxiliary loss, has not been explored in semi-supervised learning, but the method itself is borrowed from [30].

**Questions:**

I have no specific questions for now. I would like to ask the authors to point out any misunderstandings I have about the above points I have raised as weaknesses.

**Limitations:**

Its limitations are discussed in the concluding section.

It would be interesting to add a discussion of the performance of the proposed method in the presence of another often discussed realistic distribution mismatch, class distribution mismatch (open set semi-supervised learning).

---

> ### Author Rebuttal · Authors · 2023-08-09
>
> Thank you for your comments. We now answer them point by point.
>
> **W A. Problem Novelty: A discussion of several closely related problems ([a], [b] and [c]) is lacking.**
>
> Our proposed FDM-SSL problem setting is different from the previous works, we summarize the similarities and differences as follows:
>
> + **Comparing with [a]**. Our FDM-SSL and [a] both address the issue of different feature distributions for labeled and unlabeled data. The main differences between FDM-SSL and [a] lie in: **(1) test distributions.** [a] assumes that the feature distributions of test and unlabeled data are the same, diverging solely from labeled distribution. So [a] focuses on the model's performance on unlabeled distribution. Differently, our FDM-SSL framework imposes no constraints on the test distribution. The feature distribution of test data may encompass labeled, unlabeled, or even unseen ones during the training process. This presents a bigger challenge compared to [a]. **(2) cause of feature distribution mismatch.** [a] is more concerned about the different distribution of labeled and unlabeled data caused by *selection bias*, such as the construction of labeled and unlabeled data on the MNIST dataset in [a]. In contrast, our FDM-SSL is more concerned about the mismatch in feature distribution between labeled and unlabeled data caused by *image corruption* or *style changes*. Overall, there is a significant difference between FDM-SSL and [a].
> + **Comparing with [b]**. The main differences between our FDM-SSL and Semi-Supervised Domain Generalization (SSDG) are: **(1) domain labels.** SSDG assumes that the training data has $K$ domains, providing domain labels for each sample. However, the domain labels are not presented in our settings since we assume there is no additional information for unlabeled data. Consequently, FDM-SSL is a more realistic setting with fewer constraints. **(2) distributions of labeled and unlabeled data.** In SSDG, each domain contains a set of labeled data and a set of unlabeled data. These labeled and unlabeled data are sampled from the same distribution. In contrast, in our FDM-SSL, there are no constraints on the feature distribution of unlabeled data. In detail, unlabeled data could potentially include samples sharing the same feature distribution as the labeled data, or there may be no unlabeled data sharing the same distribution as labeled data.
> + **Comparing with [c]**. The main differences between our FDM-SSL and Few-Shot Domain Generalization (FSDG) are as follows: **(1) number of labeled samples.** FSDG often assumes a large number of labeled samples in source domain for training with a few labeled samples in target domain for adaptation. In contrast, in FDM-SSL, the number of labeled samples is very limited but with a large number of unlabeled samples for training. **(2) distribution shift.** FSDG focuses on generalizing a model to new classes with a limited number of label examples, which implies a shift in the label space between the test set and the training set ( $p_{test} (y)  \neq p_{train} (y)$ ). On the contrary, FDM focuses more on the robustness of feature distribution shift ( $p_ {test} (x)  \neq p_ {train} (x)$ ).
>
> **W B. Experiments: Comparisons with existing methods for the related problems I listed in Weakness A and more recent domain adaptation methods are missing.**
>
> Based on our answer to Weakness A, our FDM-SSL is different from domain generalization. Instead, it presents a more realistic SSL problem with fewer constraints. Given the mismatch in feature distribution among samples in FDM-SSL, coupled with the similarities between domain generalization and domain adaptation,  we choose the domain generalization and domain adaptation methods for comparison in our experiments.
>
> For domain generalization methods, considering that [a] does not have open-source code and [c] has different settings on the training set compared to SSL, we evaluate [b] in FDM-SSL setting.
> The following table shows the results of [b] on OFFICE-HOME dataset. We can observe that our method can achieve better performance than [b].
> In addition, [b] introduced an additional style image generation model AdaIN  to expand the training data resulting in a higher training cost (much higher memory usage and training time than our method). So our SSFA is more simple and efficient than [b].
>
> | Method| A/ACPR| |P/ACPR| |R/ACPR | |
> | ---------- | ------ | ---- | ------ | ---- | ------ | ---- |
> | |L|UL|L|UL|L|UL|
> | StyleMatch |52.0|35.3|62.9|45.9|40.3|22.3|
> | FM-SSFA|**55.0**|**45.5**|**71.8**|**52.6**|**64.8**|**52.7**|
>
> For domain adaptation, we choose the state-of-the-art UDA methods PMTrans for comparison. Detailed experiment results and analysis can be found in the response to Q3 and Q4 for Reviewer WadC.
>
> **W C. Method Novelty**
>
> For more explanation of this issue, please refer to our reply to Q1 in the general response.
>
> **Limitations. Add a discussion of the performance of the proposed method in open set semi-supervised learning.**
>
> According to the reviewer's suggestion, we conduct experiments on open set semi-supervised learning. For CIFAR10, we split the classes into known and unknown classes by defining animal classes as known (6 classes) and others as unknown (4 classes). The following table shows the accuracy.
>
> | No. of labeled samples per class |5|50|
> | :--------------------: |-----|----|
> |FixMatch|48.2|86.0|
> |FM-SSFA|**65.8**|**88.5**|
>
> In open set semi-supervised learning, our method can still bring improvements to the baseline, and the improvement is more significant when labeled data is rarer. In addition, we also conduct experiments on standard SSL benchmark or when the ratio is low (refer to the answer to Q1 from Reviewer epDa), and the results verify that our proposed method is adaptable and effective to many scenarios. To sum up, our proposed SSFA is a more general semi-supervised framework for handling inconsistent distribution of labeled and unlabeled data.

---

> > ### Comment · Reviewer_Pxab · 2023-08-16
> > **Question to authors' rebuttal**
> >
> > Thanks to the authors for their efforts in answering my questions. Some of my questions have been resolved.
> >
> > I have a question to the answer to A. Problem Novelty. Despite the fact that some past work like [a] has already considered the feature distribution mismatche between labeled and unlabeled samples, we can see that this paper contains claims that contradict this point, for example, "In this work, we introduce a novel setting to formalize the feature distribution mismatch between the labeled and unlabeled samples" in the abstract. Such statements overclaim the contributions of this paper and may mislead readers, so they should be corrected. Does the author agree with this? If yes, I would like to see a brief plan on how it could be fixed.

---

> > > ### Author Response · Authors · 2023-08-18
> > > **Answer to reviewer's question**
> > >
> > > Thank you for pointing out that some statements about FDM-SSL are not very rigorous. We first clear up the confusion and then provide our revision plan to make the description of FDM-SSL more accurate.
> > >
> > > The Covariate-shift SSL problem discussed in reference [a] assumes that the test and unlabeled feature distributions are shifted with respect to labeled feature distribution.
> > > Our FDM-SSL setting  is more general and challenging than the Covariate-shift SSL problem, designed for more complex and realistic scenario.
> > > We emphasize the differences in two aspects:
> > > + **Test distribution.** Covariate-shift SSL assumes the test data and unlabeled data are drawn from the same distribution, while FDM-SSL focuses on a more realistic scenario of different labeled, unlabeled and even unseen distributions simultaneously (see Table 1 in our paper).
> > >
> > > + **Mixed Unlabeled distribution.** In FDM-SSL, we address a scenario where unlabeled samples may come from a diverse mixture of multiple distributions, rather than a single distribution (see Section 3 in our paper).
> > >
> > > Following your suggestion, we will refine the claim of FDM-SSL in our paper by emphasizing the aforementioned points that are considered in FDM-SSL:
> > > + **Abstract Section.** We will restate the claim in lines 3-4 as follows: *In this paper, we propose a novel SSL setting that reflects a more realistic scenario. Within the novel setting, unlabeled samples are drawn from a mixed distribution that deviates from the feature distribution of labeled samples, while the test distribution covers labeled, unlabeled, and even unseen data distributions simultaneously*.
> > > + **Introduction Section.** We will restate the claim in lines 37-38 as follows: *In this study, we focus on a more realistic scenario FDM-SSL, i.e., the feature distributions of labeled and unlabeled data could be different and the feature distributions of test data could contain multiple distributions*.
> > > + **Related Work Section.** We will incorporate a comprehensive comparison between FDM-SSL and Covariate-shift SSL (as well as other related problem settings you raised) to show the differences.
> > > + **Conclusion Section.** We will restate the claim in lines 323-324 as follows: *In this paper, we focus on a realistic SSL setting, FDM-SSL, involving a mismatch between the labeled and unlabeled distributions, complex mixed unlabeled distributions and widely unknown test distributions*.
> > >
> > > If there is something you feel we have not adequately addressed yet, please do not hesitate to question.

---

> > > > ### Author Response · Authors · 2023-08-20
> > > > **A kind reminder**
> > > >
> > > > Dear Reviewer Pxab,
> > > >
> > > > We sincerely thank you for your time and efforts in reviewing our paper. We believe that we have addressed your concerns. We would appreciate it if you kindly let us know of any other concerns you may have, and if we can provide any further clarification on any other issues.
> > > >
> > > > Best wishes
> > > >
> > > > Authors

---

### Official Review · Reviewer_epDa · 2023-07-05

**Soundness:** 3 good
**Presentation:** 3 good
**Contribution:** 3 good
**Rating:** 6
**Confidence:** 4

**Summary:**

The paper challenges the traditional assumption of semi-supervised learning: the feature distributions of labeled and unlabeled data are consistent. It claims that this assumption rarely holds in realistic scenarios, as: (a) unlabeled samples could contain various corruptions; (b) unlabeled samples could contain unseen styles.

For this, the authors propose Self-Supervised Feature Adaptation (SSFA), a generic framework for improving SSL performance when labeled and unlabeled data come from different distributions.

SSFA consists of a semi-supervised learning module and a feature adaptation module, where the semi-supervised learning module performs semi-supervised and self-supervised learning

**Strengths:**

(1) The paper is well written and easy to follow, presentation is good.

(2) The proposed method is simple and effective. It consistently improves over its vanilla SSL algorithm counterpart.

(3) The paper conducts extensive experiments on two realistic scenarios and ablation study to validate the contribution of each component and the overall method.



**Weaknesses:**

(1) The proposed method has limited novelty. Incorporating self-supervised learning into semi-supervised learning, or leveraging self-supervised learning for domain adaptation, both are not new conceptions.

(2) Some important technical details are missing in the experimental setting section. For example, what is the number of shared layers and the self-supervised learning algorithm by default. These are important for reproducibility.



**Questions:**

(1) For Table 1, I am curious about the proposed method's performance on standard SSL benchmark, or when the ratio is low (like 0.1), this is a measure of whether the proposed method is robust to any scenarios, even if the noise is not that much.

(2) The ablation study shows that the number of shared layers won't affect much as long as not sharing all, I would suggest the authors to conduct experiments on more backbones and get some empirical conclusion on how to set this hyper-parameter.


**Limitations:**

Limitation of this work is that the performance of SSFA is affected by the shared parameters between the main task and the auxiliary task.
The ablation study shows that the performances do not differ much as long as not all parameters are shared.
The number of layers to share should be a hyper-parameter to tune.

---

> ### Author Rebuttal · Authors · 2023-08-09
>
> Thanks for your useful comments! We see that your main concerns are novelty, experimental details and performance. We will address your concerns.
>
> **W1. The proposed method has limited novelty. Incorporating self-supervised learning into semi-supervised learning, or leveraging self-supervised learning for domain adaptation, both are not new conceptions.**
>
>    Thank you for your comments on novelty. To our knowledge, our proposed SSFA is the first to propose to correct pseudo labels for unlabeled data through feature adaptation in SSL.  Our extensive experiments in various scenarios have also verified the effectiveness of our method. Therefore, SSFA is a concise yet effective SSL framework that can be used to solve more realistic SSL problems.
>
> Please refer to our reply to Q1 in the general response for more detailed explanations.
>
> **W2. Some important technical details are missing in the experimental setting section. For example, what is the number of shared layers and the self-supervised learning algorithm by default. These are important for reproducibility.**
>
>    Thanks for your suggestions. We will add more experimental details in the paper to enhance reproducibility. By default, the number of shared layers in our experiment is 2, the default self-supervised learning task is the rotation prediction task, and the corresponding default self-supervised loss is the cross entropy loss.
>
>
> **Q1. For Table 1, I am curious about the proposed method's performance on standard SSL benchmark, or when the ratio is low (like 0.1), this is a measure of whether the proposed method is robust to any scenarios, even if the noise is not that much.**
>
>    According to your suggestion, we conducted experiments on standard SSL setting and on the setting of low ratio(=0.1) on CIFAR100 dataset with 400 labeled data. The results are summarized in the following table. It can be seen that our method can still bring significant improvements over the baseline. This indicates that our proposed SSFA framework is robust to SSL scenarios even under a low noise ratio. The results further demonstrate that our SSFA is a generalized SSL framework.
>
>    | Method   | standard  |  SSL    | ratio  0.1 |      |      |
>    | -------- | ------------ | ---- | --------- | ---- | ---- |
>    |          | L/UL         | US   | L         | UL   | US   |
>    | FixMatch | 33.3         | 25.8 | 31.7      | 10.5 | 24.8 |
>    | FM-SSFA  | **41.3**        | **33.0** | **41.2**      | **15.8** | **32.9** |
>
> **Q2. The ablation study shows that the number of shared layers won't affect much as long as not sharing all, I would suggest the authors to conduct experiments on more backbones and get some empirical conclusion on how to set this hyper-parameter.**
>
>    Following your suggestions, we conducted relevant experiments on WiderResNet to explore the impact of the number of shared layers on CIFAR100 benchmark. In WiderResNet, "3 layers" represents that the main task and self-supervised task share the whole feature extractors. As shown in the following table,  combined with our SSFA, the baseline has a significant improvement.
>
>    In fact, in most cases, adjusting the number of shared layers does not result in significant performance fluctuations. In rare cases, sharing all feature extractors may lead to potential risks. Based on our findings, we provide a more suitable shared layer setting: sharing half of the parameters of the shared feature extractor which is able to ensure relatively better performance.
>
>    | Method               | 400  |   labeled   | 4000 |   labeled    | 4000 |  labeled     |
>    | -------------------- | ----------- | ---- | ------------ | ---- | ------------ | ---- |
>    |                      | L           | UL   | L            | UL   | L            | UL   |
>    | FixMatch             | 15.7        | 3.5  | 53.0         | 16.0 | 65.3         | 33.0 |
>    | FM-shared (2 layers) | **25.7**        | **22.2** | **60.2**         | **52.5** | **69.1**         | **57.8** |
>    | FM-shared (3 layers) | 23.3        | 21.0 | 57.3         | 47.3 | 66.2         | 53.0 |

---

### Official Review · Reviewer_wRdb · 2023-07-05

**Soundness:** 3 good
**Presentation:** 2 fair
**Contribution:** 3 good
**Rating:** 6
**Confidence:** 4

**Summary:**

In this work, the authors propose a generalized Self-Supervised Feature Adaptation (SSFA) framework for FDM-SSL (Feature Distribution Mismatched Semi-Supervised Learning) that does not require prior knowledge of the distribution of unlabeled data. The SSFA framework aims to address distribution mismatch by decoupling pseudo-label predictions from the current model. It consists of two modules: the semi-supervised learning module and the feature adaptation module. The authors draw inspiration from previous work on auxiliary tasks and incorporate an auxiliary self-supervised task into the SSL module to train alongside the main task. In the feature adaptation module, the current model, primarily trained on labeled data, is updated by utilizing the self-supervised task to adapt the feature extractor before making predictions on unlabeled data. This adaptation allows for the generation of more accurate pseudo-labels, which in turn assist the SSL process.

**Strengths:**

- Addition of the feature adaptation module to adapt the model (trained on labeled data) to the unlabeled data distribution before generating pseudo labels seems to help with the distribution shift problem between labeled and unlabeled data.

- Extensive empirical evidence provided to show that proposed SSFA method significantly improves classification accuracy on CIFAR100 and OfficeHome and Office31 datasets. Domain-level and class-level feature visualization also suggest successful feature adaptation across different domains and learning of more distinguishable representations across different classes. Ablations studies investigating the effect of different self-supervised tasks, number of shared layers between auxiliary and main task, and the sensivity of the method to different confidence thresholds are also useful.

**Weaknesses:**

- Lemma 1 is fine but convexity is a very strong assumption, which does not hold true in real-world deep loss functions. Some empirical evidence is provided, but it is somewhat counterintuitive to think that minimizing self-supervised empirical loss will also minimize supervised empirical loss.

- Using weakly augmented samples for a supervised loss and strongly augmented samples for an unsupervised loss is not novel and first introduced in the FixMatch paper. Technical novelty is somewhat incremental though feature adaptation module can still be considered a significant improvement.

**Questions:**

It was not very clear from the text about how the feature adaptation model is being trained. The module is being trained for one iteration and then is being discarded. This part is confusing. Is the module trained one unlabeled sample at a time or using all unlabeled samples? If the module needs all unlabeled samples to be trained this may not reflect a real-world use case scenario and will limit the applicability of the approach. Figure 1 needs to be updated to reflect these important details.

What kind of self-supervised loss is used in equation 5?

Please define G in Lemma 1.  What does subscript m in the loss function in Lemma 1 indicates? It was not previously used in defining loss function.

Please correct the typos in Table 1. "Traditioal",   Abundant/Scarce could replace Plenty/Lack.


-------------
Your responses have clarified some of the concerns I had with Lemma 1.  In particular, I acknowledge your explanations regarding the training process of the feature adaptation module, and the flexible nature of the self-supervised loss function. I am inclined to update my initial score to reflect these additional insights your rebuttal offered. Thanks.

**Limitations:**

Conclusions allude to some algorithmic limitations yet do not discuss societal impact. No negative societal impact beyond what is already present in generic machine learning algorithms has been observed.

---

> ### Author Rebuttal · Authors · 2023-08-09
>
> Thanks for your valuable comments! Below are the responses to your comments.
>
> **W1. Lemma 1 is fine but convexity is a very strong assumption, which does not hold true in real-world deep loss functions. Some empirical evidence is provided, but it is somewhat counterintuitive to think that minimizing self-supervised empirical loss will also minimize supervised empirical loss.**
>
> Thank you for your comments on lemma1. We will answer the questions in two parts.
>
> **1.1. It is somewhat counterintuitive to think that minimizing self-supervised empirical loss will also minimize supervised empirical loss.**
>
> Please refer to our reply to Q2 in the general response. We provide a toy example to explain this question.
>
> **1.2. Lemma 1 is fine but convexity is a very strong assumption, which does not hold true in real-world deep loss functions.**
>
> Please refer to our reply to Q3 in the general response. We provide a theoretical analysis and empirical evidence to prove its feasibility in the real-world.
>
> **W2. Using weakly augmented samples for a supervised loss and strongly augmented samples for an unsupervised loss is not novel and first introduced in the FixMatch paper. Technical novelty is somewhat incremental though feature adaptation module can still be considered a significant improvement.**
>
>    Thank you for your approval of method novelty. The consistency loss introduced by FixMatch is not the primary focus of our method. The key innovation is to propose a unified framework SSFA for FDM-SSL.  In our paper, FixMatch serves as an illustrative example to showcase the working pipeline of SSFA, but FixMatch can be replaced by any other pseudo-label-based semi-supervised method.
>
>    As your mentioned, our innovation mainly lies in the proposed feature adaptation module, which introduces a self-supervised task to achieve feature alignment. This feature adaptation module offers flexibility, allowing multiple self-supervised tasks such as rotation prediction tasks and entropy minimization tasks. Overall, our work focuses on proposing a universal SSFA framework to solve the new and more realistic FDM-SSL scenario.
>
>  Please refer to our reply to Q1 in the general response for more details.
>
>
> **Q1. It was not very clear from the text about how the feature adaptation model is being trained. The module is being trained for one iteration and then is being discarded. This part is confusing. Is the module trained one unlabeled sample at a time or using all unlabeled samples? If the module needs all unlabeled samples to be trained this may not reflect a real-world use case scenario and will limit the applicability of the approach. Figure 1 needs to be updated to reflect these important details.**
>
> We are not using all unlabeled data for updating the feature adaptation module. Instead, we only utilize the unlabeled data from the current batch. Typically, the batch size is not excessively large (e.g., set to 64 in our experiment). In particular, we update the feature extractor $\theta_ g$ to $\theta_ g'$  using the unlabeled data from the current batch. Afterwards, the updated feature extractor $\theta_ g'$ is used to extract new features for the unlabeled data to generate pseudo-labels. Notably,  $\theta_ g'$ did not have any other impact on subsequent model training. We will add important details to Figure 1 in the revised version.
>
> **Q2. What kind of self-supervised loss is used in equation 5?**
>
>    The self-supervised loss can be chosen flexibly depending on the used self-supervised task. In our paper, we employ different self-supervised losses corresponding to the rotation prediction task, contrastive learning task, and entropy minimization task, which are the cross entropy loss, the contrastive loss from SimCLR  and the entropy loss, respectively.
>
> **Q3. Please define G in Lemma 1. What does subscript m in the loss function in Lemma 1 indicates? It was not previously used in defining loss function.**
>
>    G is a constant, denoting the upper bound of $\lVert \nabla l_s(x,y;h)\rVert$. $l_m$ is the loss function of the main task.
>
> **Q4. Please correct the typos in Table 1. "Traditioal", Abundant/Scarce could replace Plenty/Lack.**
>
>    Thanks for your suggestions. We will modify the paper based on the suggestions.
>
> **Limitations. Conclusions allude to some algorithmic limitations yet do not discuss societal impact. No negative societal impact beyond what is already present in generic machine learning algorithms has been observed.**
>
> Thanks for your suggestion. We will add a section to discuss the societal impact in the paper.

---

> > ### Comment · Reviewer_wRdb · 2023-08-13
> >
> > Your responses have clarified some of the concerns I had with Lemma 1. I also acknowledge your explanations regarding the training process of the feature adaptation module, and the flexible nature of the self-supervised loss function. I have updated my initial score to reflect these additional insights your rebuttal offered. Thanks.

---

### Official Review · Reviewer_WadC · 2023-07-05

**Soundness:** 3 good
**Presentation:** 4 excellent
**Contribution:** 2 fair
**Rating:** 5
**Confidence:** 4

**Summary:**

The authors propose an interesting and novel semi-supervised learning method with auxiliary self-supervised feature adaptation for addressing issues with heterogeneity in labelled and unlabelled data. In fact, the authors focus on heterogeneity across domains, which is not something traditionally considered in the SSL literature, but is more in the field of unsupervised domain adaptation. The authors combines ideas of feature adaptation from UDA with consistency regularization and pseudo-labelling from SSL to bridge the two fields. The results presented seem much better than the baseline methods.

**Strengths:**

1. Table 1 is useful and sets the tone of the paper showing the problem settings and constraints of each type of approach.
2. The performance metrics reported in Table 2 and 3 are very impressive compared to baselines.

**Weaknesses:**

1. However, the caveat is that the baseline methods cannot do both SSL and UDA, which makes comparisons of performance metrics harder. However, SSFA does seem to be a novel method bridging the two fields.

Minor points:
1. Typo in Table 1: traditional
2. Might be useful to point out more forcefully Eq 2 is traditional in SSL, but Eq 7 is actually used, especially as SSFA Algorithm with equation references is in Supplement. Maybe it would be useful to not call both Eq 2 and Eq 7, L_u.
3. How was the cluster representation in Figure 4 made?

**Questions:**

1. The Feature Adaptation Module seems like 1-directional (unlabelled to labelled) and 1step transformation, as the unlabelled are aligned to the labelled. Have the authors tried using more steps in the optimization to allow more adaptation, or does that rely on the loss and step calibration?
2. The authors show the effect of different fixed thresholds in Eq 7 in Figure 5. Have the authors tried an adaptive threshold as in FlexMatch?
3. While the authors have compared SSFA to more recent and state-of-the-art models in SSL such as FreeMatch and SoftMatch, the authors have not compared against the arguably the state-of-the-art in UDA such PMTrans. How does SSFA compare against PMTrans or any of the more recent leaders the Domain Adaptation leaderboard on Office-31 and Office-Home?
4. Moreover, SSFA is especially impressive for low number of labelled data against UDA methods. How does it do in low labelled data settings against more recent UDA methods?

**Limitations:**

The authors have addressed some limitations of their method

---

> ### Author Rebuttal · Authors · 2023-08-09
>
> Thank you for your constructive comments! We address all of your concerns point by point as below.
>
> **W1. However, the caveat is that the baseline methods cannot do both SSL and UDA, which makes comparisons of performance metrics harder. However, SSFA does seem to be a novel method bridging the two fields.**
>
> Currently, research on FDM-SSL remains limited.   Considering the similarities between the task setting of FDM-SSL and more extensively studied SSL and UDA, we also conducted experiments on SSL and UDA methods to evaluate their ability in addressing FDM-SSL in Table 2 and Table 3 of our paper. As you pointed out, our SSFA framework has the potential to serve as a universal baseline, bridging SSL and UDA in the future.
>
> **W2. Minor points:**
>
>    1. **Typo in Table 1: traditional**
>
>        Thanks for pointing out this issue.   We will correct this typo in the final version.
>
>    2. **Might be useful to point out more forcefully Eq 2 is traditional in SSL, but Eq 7 is actually used, especially as SSFA Algorithm with equation references is in Supplement. Maybe it would be useful to not call both Eq 2 and Eq 7, L_u.**
>
>       Thanks for your suggestions. We will make modifications based on the suggestions provided by the reviewer.
>
>    3. **How was the cluster representation in Figure 4 made?**
>
>       In Figure 4, we randomly select four categories and extract features from labeled and unlabeled samples using the model's feature extractor. Then, we use t-SNE to reduce the dimensionality and visualize these features. Figure 4 (a) shows that the features of different categories are mixed together, indicating that FixMatch lacks the ability to distinguish different categories under FDM-SSL settings. In Figure 4 (b), the categories are clearly distinguished without being affected by different domains, indicating that the SSFA module can greatly alleviate the classification error caused by feature distribution mismatch.
>
>
> **Q1. Have the authors tried using more steps in the optimization to allow more adaptation, or does that rely on the loss and step calibration?**
>
>    Following your suggestion, we further explored multi-step adaptation during the optimization process on CIFAR100 benchmark (with 400 labeled data and the $ratio$ is 1.0).  As shown in the table below, only one step of adaptation can bring significant improvements over the baseline. And as the number of adaptation steps increases, the performance will be further improved, but with greater computational costs. Considering the trade-off between accuracy and computational cost, we perform one-step optimization in this paper.
>
>    | Method             | L     | UL    | US    |
>    | ------------------ | ----- | ----- | ----- |
>    | FM                 | 15.7  | 3.5   | 8.5   |
>    | FM-SSFA (1 step)   | 25.7  | **22.2**  | 22.5  |
>    | FM-SSFA (5 steps)  | 26.2  | 14.9  | 16.0 |
>    | FM-SSFA (10 steps) | **41.1** | 18.7 | **33.7** |
>
> **Q2. Have the authors tried an adaptive threshold as in FlexMatch?**
>
> In our paper, we have compared SSFA to more recent and state-of-the-art SSL algorithms such as FreeMatch and SoftMatch. Both algorithms use self-adaptive thresholds similar to (but more complex than) FlexMatch. For further explanation,  we conduct FlexMatch in FDM-SSL and give the results on CIFAR100 benchmark (with 400 labeled data and the $ratio$ is 0.5) as follows. It can be seen that SSFA can further improve the performance of FlexMatch, especially in the distribution of unlabeled and unseen data.
>
>    | Method         | L    | UL   | US   |
>    | -------------- | ---- | ---- | ---- |
>    | FlexMatch      | 35.8 | 2.2  | 23.6 |
>    | FlexMatch-SSFA | **38.3** |**28.4** | **31.6** |
>
> **Q3. How does SSFA compare against PMTrans or any of the more recent leaders the Domain Adaptation leaderboard on Office-31 and Office-Home?**
>
> Following your suggestion, we evaluate PMTrans under FDM-SSL setting on OFFICE-HOME benchmark. As shown in the table below, PMTrans achieves very poor performance. In fact, we have also found that previous UDA methods (such as DANN and CDAN) may crash during training in some cases. We have mentioned the relevant reasons in the paper (lines 244 - 248). In addition, due to the fact that the PMTrans uses a more complex Swin-Transformer as the backbone, while our method uses Resnet50 as the backbone, PMTrans requires longer training time and more memory usage compared to SSFA.
>
>    | Method   | A/ACPR |      | C/ACPR |      | P/ACPR |      | R/ACPR |      |
>    | -------- | ------ | ---- | ------ | ---- | ------ | ---- | ------ | ---- |
>    |          | L      | UL   | L      | UL   | L      | UL   | L      | UL   |
>    | PMTrans  | 8.6    | 2.3  | 17.4   | 10.5 | 23.8   | 12.7 | 15.3   | 6.6  |
>    | FixMatch | 32.4   | 23.0 | 36.9   | 30.6 | 52.9   | 32.6 | 42.2   | 31.5 |
>    | FM-SSFA  | **55.0**   | **45.5** | **44.7**   | **41.7** |**71.8**   | **52.6** | **64.8**   | **52.7** |
>
> **Q4. Moreover, SSFA is especially impressive for low number of labelled data against UDA methods. How does it do in low labelled data settings against more recent UDA methods?**
>
> Following your suggestion, we evaluate the recent UDA method PMTrans and our SSFA in low labeled data settings (1 labeled sample per class) on OFFICE-HOME benchmark. It can be seen that in low labeled data settings, the performances of PMTrans and FixMatch are poor. However, with our SSFA, FixMatch has a significant improvement in performance.
>
>    | Method   | A/ACPR |      | C/ACPR |      | P/ACPR |      |
>    | -------- | ------ | ---- | ------ | ---- | ------ | ---- |
>    |          | L      | UL   | L      | UL   | L      | UL   |
>    | PMTrans  | 5.8    | 1.7  | 6.6    | 3.3  | 10.4   | 6.3  |
>    | FixMatch | 4.7    | 3.5  | 10.0   | 6.6  | 6.3    | 5.7  |
>    | FM-SSFA  | **19.1**   | **20.5** | **18.4**   | **16.2** | **29.3**   | **20.0** |

---

> > ### Comment · Reviewer_WadC · 2023-08-16
> >
> > I thank the authors for clarifying some of questions and performing new experiments on state-of-the-art UDA methods. Those UDA methods were confusing, and the authors also mention that other UDA methods such DANN and CDAN can crash during training. This makes it hard to compare the method to UDA methods. While this may be no fault of the method, it does makes the method look like an interesting application of self-supervised feature to semi-supervised learning with some improved performance results, but not quite comparable to UDA methods.

---

> > > ### Author Response · Authors · 2023-08-18
> > > **Thank you for the reply!**
> > >
> > > Thank you for the reply. We analyze the reason for poor performance of UDA methods on FDM-SSL as follows:
> > >
> > > 1. UDA methods usually assume abundant source domain data for model training. However, in FDM-SSL, the labeled data is very scarce, which can potentially hinder the training process. For example, many UDA methods incorporate a domain discriminator to distinguish source and target samples. The severe class imbalance between limited labeled samples and abundant unlabeled samples poses challenges to the effective training of domain discriminators.
> > >
> > > 2. UDA typically deals with unlabeled data from a single distribution, focusing on adapting to one target distribution. In our FDM-SSL, unlabeled data can originate from multiple distributions without prior information. When employing UDA methods, the unlabeled samples are usually considered to come from a single distribution (target domain). This simplification may disturb the training of UDA methods and cause crashes when unlabeled data come from multiple distributions.  Our SSFA takes a different approach. It leverages the feature adaptation module to accommodate different distributions of unlabeled data. This enables SSFA to flexibly extract features for unlabeled data from different distributions.
> > >
> > > If there is something you feel we have not adequately addressed yet, please do not hesitate to question.

---

### Official Review · Reviewer_LyxH · 2023-07-07

**Soundness:** 3 good
**Presentation:** 4 excellent
**Contribution:** 2 fair
**Rating:** 5
**Confidence:** 4

**Summary:**

This paper focuses on Feature Distribution Mismatch SSL (FDM-SSL), which considers a mismatch between the labeled and unlabeled data. In this task setting, the challenges include the scarcity of labeled data and the mixed distributions of the unlabeled data. The paper proposes a framework named Self-Supervised Feature Adaptation (SSFA), which adapts the feature extractor to the unlabeled distribution. Experiments show that SSFA can improve SSL performance on labeled, unlabeled, and unseen data.

**Strengths:**

1. The paper studies a new SSL problem named Feature Distribution Mismatch SSL. The presented SSFA method is simple and effective to address this problem.

2. Extensive experiments have been conducted to evaluate the performance of SSFA.

3. The paper is well written and presented. For instance, Table 2 is clear, it reveals the difference between task settings.

**Weaknesses:**

1. The proposed SSFA method is built upon existing SSL and unsupervised domain adaptation (UDA) methods. In SSFA, the SSL module and Feature Adaptation Module are inherited from the benchmark SSL and UDA methods, respectively. The optimization objectives $L_x$, $L_u$, and $L_{aux}$ are widely used in SSL methods. The major novelty is the Feature Adaptation Module, which uses the unlabeled data as input to generate pseudo labels $q_b'$ instead of the original pseudo labels $q_b$ generated from the labeled data. I acknowledge that this work presents some novelties in proposing and addressing the challenges in domain-shifted SSL. However, the method itself is straightforward in my opinion.

2. The assumptions of Lemma 1 are somewhat strong, i.e., the loss is expected to be convex and smooth, the learning rate is fixed to $\frac{\epsilon}{\beta G^2}$. In addition, the result of Lemma 1 is counter-intuitive --- 'The empirical risk of the main task can be theoretically reduced to 0 by optimizing the empirical risk of self-supervised task.' Due to these aspects, I kindly doubt that this theoretical result could work well in practice. And, the derivation only guarantees that the empirical risk is decreasing along with gradient descent, but not 'reduced to 0'. Please check this.

3. I am a bit confused about the SSFA's theoretical performance on 'unseen data'. In Sections Introduction and Experiments, the authors claimed that SSFA can significantly improve SSL performance on unseen data. However, the method only adapts models to the *unlabeled* data during training. Why does SSFA improve the performance on unseen data? More theoretical evidence is encouraged to be presented to better support this claim.

4. I suggest the authors show some visualizations of the gap between labeled and unlabeled data, e.g., what are the image corruptions in the experiments? What's the style change from OFFICE-31 dataset to OFFICE-HOME dataset? I find some related details in Appendix's texts. However, it would be more friendly to show some visualizations of the data gap, since domain shifting is the main challenge studied by this work.


**Questions:**

See Weaknesses.

---

> ### Author Rebuttal · Authors · 2023-08-09
>
> Thank you for your valuable comments! We see that your main concerns are novelty, more explanations and some details. We answer your questions point by point.
>
> **W1. I acknowledge that this work presents some novelties in proposing and addressing the challenges in domain-shifted SSL. However, the method itself is straightforward in my opinion. However, the method itself is straightforward in my opinion.**
>
> Thanks for acknowledging the novelty of our work. We believe that a straightforward method offers several advantages, including simplified implementations, facilitating understanding, and serving as a solid baseline for further research in Feature Distribution Mismatch SSL.
>
> Please refer to our reply to Q1 in the global response for more details.
>
> **W2.   The result of Lemma 1 is counter-intuitive. The assumptions of Lemma 1 are somewhat strong and I kindly doubt that this theoretical result could work well in practice. Lemma 1 only guarantees that the empirical risk is decreasing along with gradient descent, but not 'reduced to 0'.**
>
> Thank you for your feedback on issues related to Lemma 1. We will divide the answer to W2 into three parts.
>
>  **2.1.The results of Lemma 1 is counter-intuitive --- 'The empirical risk of the main task can be theoretically reduced to 0 by optimizing the empirical risk of self-supervised task'.**
>
> Please refer to our reply to Q2 in the general response.
>
> **2.2 The assumptions of Lemma 1 are somewhat strong, potentially impeding the effectiveness of the theoretical result in practice.**
>
> Please refer to our reply to Q3 in the general response.
>
> **2.3 Lemma 1 only guarantees that the empirical risk is decreasing along with gradient descent, but not 'reduced to 0'.**
>
> According to the toy example, it is possible to indirectly reduce the main task loss to 0 by minimizing self-supervised task loss. In addition, according to Lemma 1, assuming that the loss is expected to be convex and smooth, as the gradient of self-supervised task loss decreases, the empirical risk of the main task will gradually decrease. Therefore, if we choose a proper training strategy to optimize the self-supervised task, the empirical risk of the main task can tend to 0.
>
> **W3. However, the method only adapts models to the unlabeled data during training. Why does SSFA improve the performance on unseen data?**
>
> We explain the reason why SSFA improves the performance on unseen data from two aspects.
>
>    + **From a perspective of method.** In our SSFA, unlabeled samples are aligned with the feature distribution of labeled samples through the feature adaptation module, thereby improving the model's prediction accuracy for distribution-mismatch unlabeled samples. By making correct predictions on samples from various domains, the feature extractor becomes adept at mapping data from different domains to the same feature space, gradually eliminating the impact of domain divergence.  As a result, the model learns to capture domain-invariant features as two feature space becomes indistinguishable.
>
>    + **From a perspective of experiment.** As the domain number exposed by the model increases during training, the feature extractor tends to map more domains to the same feature space. It enhances the model's potential for generalizing to unseen domains. Therefore, the model has a higher improvement compared to the baseline on unseen data. To visually illustrate this, we visualize the domain-level features generated by SSL models with/without SSFA respectively. As Figure 1 in the PDF of the global response shows, FixMatch maps samples from different domains (1 labeled, 10  unlabeled and 5 unseen domains) to different clusters in the feature spaces. Differently, our FM-SSFA model can effectively fuse these samples. The fusion of features in the unseen domains and the seen domains indicates that our SSFA has a good generalization ability to map different domain features to the same feature space.
>
> **W4. I suggest the authors show some visualizations of the gap between labeled and unlabeled data, e.g., what are the image corruptions in the experiments?**
>
>    Thanks for your suggestions. We will add relevant visualization images in our paper to further illustrate the feature distribution gap between labeled and unlabeled data. We show the visualizations of the gap caused by image corruption and style change in Figure 2 and Figure 3 in the PDF of the global response respectively.

---

### Author Rebuttal · Authors · 2023-08-09

We are grateful to all reviewers and ACs for the generous effort they have invested in reviewing our work! Some important or common questions are answered.

**Q1. The novelty of the method may be limited.**

Thanks for your comments.

According to Occam's Razor, *Entities should not be multiple unnecessarily*. In addition, complex methods often incur increased training costs. Therefore, our method is intentionally designed to be simple yet highly effective.

In our paper, we aim to alleviate the severe performance degradation caused by feature distribution mismatch between labeled and unlabeled data in SSL. We propose Self-Supervised Feature Adaptation (SSFA) from the perspective of distribution adaptation. By using a feature adaptation module to update the features of unlabeled samples, we can map input data with different feature distributions to the same feature space, thereby greatly improving the pseudo-labels accuracy for unlabeled samples. In this way, our proposed SSFA is not a specific semi-supervised method, but a universal semi-supervised framework based on feature adaptation.
To our knowledge, our proposed SSFA is the first to propose to predict pseudo labels for unlabeled data after feature adaptation in SSL.  This method is characterized by its simplicity and remarkable effectiveness in solving more complex SSL problems.
Furthermore, in experiments, we have demonstrated that the SSFA framework can be combined with a wide range of pseudo-label-based semi-supervised learning methods. Besides, there is a wide selection of auxiliary tasks for feature adaptation modules, which greatly improves the practicality of our proposed SSFA framework. We believe that this simple framework can serve as a solid baseline for further research in Feature Distribution Mismatch SSL.

**Q2. The results of Lemma 1 is counter-intuitive.**

We provide a toy example to illustrate this counter-intuitive conclusion pointed by the reviewer: minimizing the loss of self-supervised tasks in Lemma1 indirectly minimizes that of the main task.

Consider a two-layer linear network parametrized by $\theta_g$ (the shared linear layer) , $\theta_c$ (the linear head for main task) and $\theta_s$ (the linear head for self-supervised task) . The predictions for the main and self-supervised task can be denoted as: $\hat y_m = \theta_c^T \theta_g x$ and $\hat y_s = \theta_s^T \theta_g x$ separately. The main task loss is $l_m(x,y; \theta_g, \theta_c)=\frac12(y_m-\hat y_m)^2$ and the self-supervised task loss is $l_s(x,y; \theta_g, \theta_s)=\frac12(y_s-\hat y_s)^2$. Since $y_s$ is known, we update the shared feature extractor $\theta_g$ by one step of gradient descent on $l_s$. The updated feature extractor $\theta_g'$ is given by:
        $$
        \theta_g'= \theta_g - \eta(y_s-\hat y_s)(-\theta_sx^T)= \theta_g - \eta(y_s-\theta_s^T \theta_g x)(-\theta_sx^T),
        $$
        where $\eta$ is the learning rate. If we set $\eta=\frac{y_m-\hat y_m}{(y_s-\hat y_s)\theta_c^T\theta_sx^Tx}$, we can find the main task loss $l_m(x,y; \theta_g', \theta_c)=0$.

This toy example demonstrates that it is theoretically possible and reasonable to reduce the loss of the main task to zero, by optimizing self-supervised tasks.


**Q3. The assumptions of Lemma 1 are somewhat strong, which may not hold true in practice.**

Although Lemma 1 relies on somewhat strong assumptions, its conclusion can be applied to real-world scenarios where the assumptions might not hold.
We attempt to verify this point by first offering a theoretical analysis and then providing empirical evidence derived from the theory.

+ **Theoretical Analysis**: Based on lemma1, for any $\eta$,  by smoothness and convexity,
$$l_m(x,y;h-\nabla l_s(x;h))\leq l_m(x,y;h)+\eta{\langle \nabla l_m(x,y;h),\nabla l_s(x;h)\rangle}+\frac{\eta^2 \beta}2 {\lVert \nabla l_s(x;h)\rVert}^2.$$
Denote $\eta^*=\frac{\langle \nabla l_m(x,y;h),\nabla l_s(x;h)\rangle}{\beta \lVert \nabla l_s(x;h)\rVert^2}$.
The equation becomes:
$$l_m(x,y;h-\eta^*\nabla l_s(x;h))\leq l_m(x,y;h)-\frac{\langle \nabla l_m(x,y;h),\nabla l_s(x;h)\rangle^2}{2\beta \lVert \nabla l_s(x;h)\rVert},$$
 namely,
$$ l_m(x,y;h)-l_m(x,y;h-\eta^*\nabla l_s(x;h))\geq \frac{\langle \nabla l_m(x,y;h),\nabla l_s(x;h)\rangle^2}{2\beta \lVert \nabla l_s(x;h)\rVert}.$$
It can be seen that in the smooth and convex case, with the gradient inner product between the main task loss $l_m$ and the self-supervised task loss $l_s$, i.e., $\langle \nabla l_m(x,y;h),\nabla l_s(x;h)\rangle$, increasing, the updated model has a smaller loss on the main task. That is, the larger the inner product, the greater the decline of the loss function.

+ **Empirical Evidence**: For non-convex loss functions, we empirically show that our theoretical insights also hold. In Figure 2 of our paper, we plotted the correlation between the gradient inner product and the performance improvement of the model on the test set, where each point in the figure represents the average result of a set of test samples. From Figure 2, it can be seen that there is a positive correlation between the gradient inner product and model performance improvement. This observed phenomenon is consistent with the theoretical conclusion, that is, a strong gradient correlation clearly indicates higher performance improvements over the baseline.

---

### Decision · Program_Chairs · 2023-09-21

**Decision:**

Accept (poster)

**Comment:**

The paper studies an extension of the semi-supervised learning (SSL) setting in which feature distributions of labeled and unlabeled data can be different. The challenge when applying standard SSL methods in this setting is that the pseudo-labels tend to be incorrect due to the feature distribution mismatch. The authors propose a method for solving this problem by decoupling pseudo-label predictions from the current model. The proposed method consists of two modules: SSL learning module and feature adaptation module. SSL module is based on auxiliary self-supervised task inspired by previous works. Feature adaptation module is designed to produce more reliable samples by adapting the model to the unlabeled data distribution before making predictions. The method is evaluated with image corruption and style change as distribution mismatch scenarios.

Reviewers found the proposed method to be simple and effective. The paper is well-written, well-organized and easy to follow. The experimental evaluation is comprehensive and the improvements over baselines are substantial. The ablation studies are carefully designed and validate the proposed approach.

Some reviewers expressed concerns about (1) the problem's novelty and lack of discussion and proper positioning with respect to prior works that considered related problems, (2) the usefulness of theoretical results in reality given strong assumptions, and (3) lack of comparison to more recent UDA methods. During the rebuttal, these concerns have been addressed by the authors and reviewers raised their scores.

Eventually all reviewers agree that the paper should be accepted and thus my recommendation is to accept the paper.